# Complementarity Matters: A Closer Look at Nearest Neighbor Guidance for OOD Detection

## Abstract

Out-of-distribution (OOD) detection seeks to identify inputs that fall outside the training distribution, which is crucial for ensuring the reliability of deep neural networks (DNNs). The dominant approach to OOD detection is score-based: each sample receives a score, and those with scores that differ significantly from the training data are flagged as out-of-distribution. The most effective score functions typically rely on DNN's classifier uncertainty, or nearest-neighbour (NN) similarity of test and training samples. Moreover, recent research demonstrates that combining the classifier- and NN-based scores - the process called NN Guidance - yields the best OOD detectors. However, the exact reasons for the success of NN Guidance are poorly understood. In this work, we take a closer look at NN Guidance and uncover the core reason behind its success - the complementarity between the classifier- and NN-based scores. Put simply, the two scores are complementary when they detect diverse OOD samples, and thus they can perform better when combined. Guided by these insights, we make three main contributions. First, we design a strong baseline OOD detector based on NN Guidance with improved score complementarity. Second, we propose a novel model pruning strategy that further enhances the complementarity and improves performance. Third, we propose a novel method to combine complementary signals from different hidden DNN layers to improve NN Guidance. Finally, by integrating all the sources of complementary information into a unified framework - CoNNGuide - we achieve state-of-the-art performance on CIFAR and ImageNet benchmarks, outperforming prior methods by up to 5.7% in FPR and 2.79% in AUROC.

## 1 Introduction

Modern machine learning systems are increasingly deployed in open-world settings, where inputs encountered at test time may differ significantly from the training distribution Afshari et al. (2022); Hendrycks et al. (2021). Such out-of-distribution (OOD) inputs can cause models to fail silently, making OOD detection a critical safety requirement in applications like autonomous driving and medical diagnosisHains et al. (2018); Ren et al. (2021); Zadorozhny et al. (2023). A growing body of research has focused on deriving score functions from neural networks to distinguish in-distribution (ID) from OOD inputs to address this challenge.

The dominant approach to OOD detection involves deriving a scalar score function that captures the "familiarity" of an input—assigning high scores to in-distribution (ID) data and low scores to unfamiliar, OOD examples. Two major families of methods have emerged: classifier-based scores, which use a model's predictive confidence (e.g., softmax, energyLiu et al. (2020)), and distance-based scores, which assess similarity to training samples in feature space via k-nearest neighbors (KNN) Sun et al. (2022). Classifier-based methods excel at capturing fine-grained differences near decision boundaries, but often fail far from the data manifold. Conversely, distance-based methods are more reliable for far-OOD detection but struggle with near-OOD inputs.

A promising recent direction is nearest-neighbor (NN) guidance, which aims to combine the strengths of classifier- and NN-based scores. One such method, NNGuide Park et al. (2023), computes a unified score that leverages both components. While NNGuide achieves strong empirical

performance when paired with certain classifier-based scores, its effectiveness varies widely depending on the score it guides—sometimes offering no gain. The factors causing this variance remain poorly understood.

In this work, we revisit NNGuide from a principled perspective and ask: what exactly enables improved OOD detection? Our investigation reveals two key factors: The first is the complementarity between the classifier-based and nearest-neighbor scores. To put it simply, if the two scores, when applied independently, detect diverse OOD examples, then combining them via guidance will give a larger improvement. The second factor is the feature space used for nearest neighbor search. Different DNN hidden layers produce features with different properties, thus affecting the nearest neighbor search and consequently the guidance quality.

With access to the above insights, we propose a principled way to build a high-performance OOD pipeline using NN guidance. First, we construct a strong baseline OOD detector by combining existing improvements to the classifier-based score, and show that higher classifier- and NN-based scores complementarity reflects the improved performance. Second, we propose a novel weight pruning strategy that improves the robustness of classifier-based scores, leading to improved complementarity with the NN score and better OOD performance when used with NN guidance. Third, we propose multi-guidance - a way to combine multiple NN scores (computed at different DNN layers) into one, which further promotes complementarity with classifier-based scores and improves OOD detection. Finally, we build the whole pipeline - CoNNGuide (Complementary Nearest Neighbor Guidance) - which integrates the complementary contributions of each component and achieves state-of-the-art OOD detection performance.

To evaluate the performance of CoNNGuide, we perform experiments for OOD detection on three popular benchmarks, namely, CIFAR-10, CIFAR-100, and ImageNet with the DenseNetHuang et al. (2017), ResNet50 He et al. (2015), and RegNet Xu et al. (2023) models. Our results show that CoNNGuide outperforms prior methods by up to 5.7% in false positive rate (FPR) and 2.79% in AUROC.

In summary, this work makes the following contributions: 1) **A principled framework for analyzing NN Guidance with score complementarity**, leading to a strong baseline OOD detector composed of carefully matched classifiers- and NN-based scores. 2) **A novel network pruning strategy and new score guidance formulation**, both of which enhance score complementarity and improve NN guidance. 3) **A unified OOD detection pipeline, CoNNGuide**, that achieves state-of-the-art performance across multiple benchmarks and model architectures.

The rest of the paper is organized as follows: Section "Related work" compares CoNNGuide to related work. Section "Preliminaries" gives the background material for our work. Section "Our Approach" describes our methodology, CoNNGuide, to leverage existing OOD detection approaches to filter effectively the unreliable DNN outputs. Section "Experimental results" presents the empirical evaluation of the CoNNGuide's effectiveness and efficiency. Section "Conclusion" concludes the paper and discusses future research.

## 2 RELATED WORK

Out-of-distribution (OOD) detection has emerged as a critical component for building reliable machine learning systems. Broadly, prior works can be categorized into two methodological lines: classifier-based approaches and distance-based approaches. Classifier-based methods operate on a trained model's output confidence, often using scores such as the maximum softmax probability (MSP), energy Hendrycks & Gimpel (2018), or logit-based alternatives like MaxLogit Hendrycks et al. (2022) and KL divergence Hendrycks et al. (2022). These approaches are effective at capturing uncertainty near decision boundaries but tend to misclassify far-OOD examples overconfidently.

Distance-based methods, on the other hand, evaluate the similarity between test inputs and training data in feature space. Techniques such as the Mahalanobis distance Lee et al. (2018), SSD Sehwag et al. (2021), and KNN-based detection Sun et al. (2022) have shown strong performance in identifying far-OOD samples. However, they often lack the fine-grained discriminative capability of classifier-based scores and underperform on near-OOD instances.

To harness the complementary strengths of these paradigms, recent work has explored hybrid methods. NNGuide Park et al. (2023) represents a seminal effort in this direction. It proposes a confidence-guided NN scoring mechanism, where classifier scores are modulated by similarity to nearby training features. This simple, post-hoc mechanism improves both robustness to far-OODs and sensitivity to near-OODs. However, the effectiveness of NNGuide is tightly coupled with the base classifier score and the underlying feature space, and its performance gains vary significantly across settings.

Beyond detection scores, various network truncation and feature rectification techniques such as ReAct Sun et al. (2021), DICE Sun & Li (2022), LINe Ahn et al. (2023), RankFeat Song et al. (2022), and BATS Zhu et al. (2022) have been proposed to enhance the ID-OOD separability. These methods often improve performance when combined with appropriate OOD scores.

## 3 PRELIMINARIES

In this section, we define the problem of OOD detection more formally and revise the details of the NN guidance Park et al. (2023) and importance-based pruning Ahn et al. (2023) approaches.

**Out-of-Distribution (OOD) Detection.** Out-of-distribution (OOD) detection identifies inputs that lie outside the training distribution, which can otherwise cause incorrect or overconfident predictions. A common approach is to compute a scalar OOD score $S(\mathbf{x})$ that reflects the model's confidence in a given input $\mathbf{x}$. If the score exceeds a threshold $\lambda$, the input is considered in-distribution (ID); otherwise, it is deemed out-of-distribution (OOD). This decision rule is formalized as:

$$D(\mathbf{x}) = \begin{cases} \text{ID}, & \text{if } S(\mathbf{x}) \geq \lambda \\ \text{OOD}, & \text{otherwise} \end{cases} \tag{1}$$

Typically, the $\lambda$ parameter is chosen to be the 95th bottom percentile of the score $S(x)$ distribution in the training set.

Detection scores can broadly be categorized into two types:

- **Classifier-based scores**, derived from the model's output logits (e.g., maximum softmax probabilityHendrycks et al. (2022), energy-based Liu et al. (2020) scores), leverage class-specific information and are typically effective at detecting near-OOD instances close to decision boundaries.

- **Distance-based scores**, such as k-nearest neighbors (KNN) Sun et al. (2022) or Mahalanobis distance Lee et al. (2018), rely on the similarity between test and training features, and are better suited to detecting far-OOD samples.

**Nearest Neighbor Guidance (NNGUIDE).** NNGuide Park et al. (2023) is a post-hoc, training-free method that combines the strengths of classifier- and distance-based OOD detection. It starts with a base confidence score $S_{\text{base}}(\mathbf{x})$ - typically classifier-derived - and adjusts it using a nearest-neighbor-based guidance term $G(\mathbf{x})$, yielding a final guided score:

$$S_{\text{NNGuide}}(\mathbf{x}) = S_{\text{base}}(\mathbf{x}) \cdot G(\mathbf{x})$$

The guidance term $G(\mathbf{x})$ is computed from a small "bank" of in-distribution samples. It reflects how similar the test input is to the top-$k$ nearest neighbors in this bank, using cosine similarity weighted by the base confidence of each neighbor. Formally:

$$G(\mathbf{x}) = \frac{1}{k} \sum_{i=1}^{k} s^{(i)} \cdot \cos(z^{(i)}, z)$$

Here, $z$ is the normalized feature of $\mathbf{x}$, $z^{(i)}$ are the top-$k$ neighbors in the training set feature space, and $s^{(i)} = S_{\text{base}}(\mathbf{x}^{(i)})$ are their base confidences.

This guidance corrects the overconfidence of classifier scores in far-OOD regions and retains their fine-grained sensitivity near ID boundaries. As a result, NNGuide can combine the strengths of classifier- and distance-based approaches and improve detection across both near- and far-OOD regimes, offering robust performance with minimal computational overhead.

**Importance-based Pruning.** Activation and weight pruning for OOD detection were first introduced as a part of the LINe method Ahn et al. (2023). It aims to reduce the effect of noisy weights and activations on the classifier-based scores, which helps to improve OOD detection. The approach identifies important neurons by computing class-wise Shapley values, a general tool used in global sensitivity analysis Da Veiga et al. (2021), and selectively retains only the most contributive ones.

Given a pre-trained model with penultimate layer activations $h(x) = [a_1, a_2, \ldots, a_d] \in \mathbb{R}^d$ for an input $x$, where $d$ is the number of neurons in the penultimate layer, and $f_\theta(x) = \left( \mathbf{W}^\top \mathbf{h}(x) + \mathbf{b} \right) \in \mathbb{R}^L$ is the class logits of $L$ classes, LINe estimates the contribution $s_i^{(l)}$ of neuron $i$ to class $l$ using a first-order Taylor approximation of the Shapley indices Shapley (1953):

$$s_i^{(l)} = \left| a_i \cdot \frac{\partial f_\theta(x^{(l)})}{\partial a_i} \right| \tag{2}$$

where $a_i$ is the activation of neuron $i$, and $x^{(l)}$ is an input from class $l$. Averaging these values over training samples of class $l$ produces a contribution matrix $\mathbf{C} \in \mathbb{R}^{d \times L}$, where each column corresponds to a class.

For each class $l$, the top-$k$ neurons with the highest contributions are selected to form a binary mask vector $\mathbf{m}^{(l)} \in \{0, 1\}^d$, where ones indicate the retained neurons. To prune the weight matrix, the authors select the top-$k$ weight entries according to the contribution matrix $\mathbf{C}$. The resulting pruned weight matrix $\hat{W}$ is then used to obtain the logits. At inference time, given a predicted class $l$, the activation-pruned model output becomes:

$$f_{\text{IP}}(x) = \hat{\mathbf{W}}^\top \left( \mathbf{m}^{(l)} \odot \mathbf{h}(x) \right) + \mathbf{b} \tag{3}$$

where $\hat{\mathbf{W}} \in \mathbb{R}^{d \times L}$ is the prunned weight matrix, $\mathbf{b} \in \mathbb{R}^L$ is the bias, and $\odot$ denotes element-wise multiplication. More details about $\mathbf{C}$, $\hat{W}$, and $m^{(l)}$ can be found in Ahn et al. (2023).

This pruning removes low-contribution neurons that may introduce noise, enabling more robust OOD detection by focusing only on class-specific informative features and weights.

## 4 EXPLAINING NNGUIDE'S PERFORMANCE THROUGH SCORE COMPLEMENTARITY

In this section, we start by sharing our findings regarding what makes NNGuide work and how to use these findings to build a SoTA pipeline for OOD detection.

We begin by taking a closer look at NNGuide, which aims to improve OOD detection by combining classifier- and distance-based scores. While the original paper reports consistent gains when classifier scores are guided by a NN-based score, not all classifier scores benefit equally. This variability is not addressed in the original work, yet we believe it is central to unlocking NNGuide's full potential.

Our hypothesis is that for NN guidance to be effective, the classifier score must be *complementary* to the NN score. That is, each score should be able to detect different OOD examples when applied independently (following the decision rule in Equation 1). To test this, we evaluate a range of classifier-based scores and measure their complementarity with a NN-based score computed from penultimate-layer features, as in Park et al. (2023).

We define complementarity as the fraction of OOD examples detected by either method alone:

$$r_{\text{comp}} = \frac{\sum_{i \in S_{\text{OOD}}} \left( I_{\text{cls}}(i) \vee I_{\text{nn}}(i) \right)}{N_{\text{OOD}}} \tag{4}$$

Here, $I_{\text{cls}}(i)$ and $I_{\text{nn}}(i)$ are indicator functions that equal 1 if the $i$-th OOD example is detected by the classifier- or NN-based method, respectively. $S_{\text{OOD}}$ denotes the set of OOD examples, with a total size of $N_{\text{OOD}}$.

This metric captures the intuition that if either score can detect an OOD sample independently, then their combination via NNGuide is likely to detect it as well. In other words, a higher complementarity ratio means a better OOD detection performance after applying NNGuide. Moreover, it has

been noticed in computer vision tasks as well that complementary information boosts NN's performance Dvornik et al. (2019), and we expect similar phenomena to occur in OOD detection. We use the complementarity ratio metric to shed light on why various modifications to the classifier-based score lead to improved OOD detection when used with NN guidance.

**NNGuide++.** We observe that the literature on OOD detection is rich with diverse scoring methods and neural network enhancements—each effective in isolation. In this section, we propose NNGuide++ - a new composite framework, based on NNGuide, with a new, improved classifier-based score. The new classifier score is obtained by carefully integrating multiple OOD detection components, such as the energy score Liu et al. (2020), ReAct Sun et al. (2021), and significance pruning Ahn et al. (2023). We show that the resulting pipeline already achieves new SoTA, and every individual component improves the overall performance. To understand why, we perform analysis through complementary ratios (see Eq 4). More precisely, starting with the energy score as the base classifier score, we add the improvements one at a time, and measure the OOD performance and the complementarity score (with an NN score). The results are shown in Table 1. Crucially, we can see that the improved OOD detection and complementarity are correlated. To verify this claim, we plug in a less capable device MaxLogit Hendrycks et al. (2022) or MSP Hendrycks & Gimpel (2018) into the pipeline and see both the improvement and the complementarity ratio go down, suggesting that the improved performance of the NN guide framework is indeed explained by the classifier- and NN-based score complementarity. The above justifies our choice of individual parts of NNGuide++ and allows for a deeper understanding of further pipeline improvements.

| Base scores | Complementarity Ratios (%) | More Complementary? | FPR95 ↓ | AUROC ↑ | Improvement |
|---|---|---|---|---|---|
| Energy Liu et al. (2020) | 89.26 | - | 24.79 | 94.99 | - |
| MaxLogit Hendrycks & Gimpel (2018) | 89.1 | x | 25.12 | 94.9 | x |
| MSP Hendrycks et al. (2022) | 86.15 | x | 45.57 | 93.79 | x |
| ReAct Sun et al. (2021) | 89.49 | ✓ | 21.21 | 96.13 | ✓ |
| ReAct+Shap Ahn et al. (2023) | 91.35 | ✓ | 13.42 | 97.4 | ✓ |

Table 1: How complementarity ratios relate to NNGuide's improvement. The first column lists the method added to NNGuide; the second, the complementarity ratio between classifier and NN-based scores; the third, whether this ratio improves over Energy only ("-" meaning not applicable). Columns four and five show OOD performance on CIFAR10, and the last column indicates improvement over Energy. "Shap" denotes activation and weight pruning from Ahn et al. (2023). The OOD score used in ReAct is Energy.

## 5 OUR APPROACH

In this section, we present our novel feature pruning strategy and a multi-guidance mechanism that further enhances the framework, which together lead to our complete method - CoNNGuide.

### 5.1 IMPORTANCE-BASED ACTIVATION PRUNING WITH CLASS–PROBABILITY INFORMATION

We focus now on improving the individual components of the pipeline, starting with model pruning based on Shapley indices. Activation pruning in OOD detection aims to improve the classifier score by removing activations that are unimportant for classification and that introduce noise into the score computation Ahn et al. (2023). While the original approach measures each neuron's contribution to the logits, classification ultimately depends on the softmax probabilities rather than the raw logits. Computing importance solely on per-class logits assumes that class contributions are independent, which can misidentify unimportant neurons that affect all logits equally (and thus do not change class probability) as important. For example, a neuron $a_i$ may change every logit by the same amount, leaving the class probabilities unchanged and thus irrelevant to the final decision. To address this undesired property, we propose *P-Shapley* indices that consider the class probabilities when computing the neuron's importance. More formally, we change the Equation 2 as follows:

$$s_i^{(l)} = \left| a_i \cdot \frac{\partial \left( f_\theta(x^{(l)}) \cdot p_\theta(x^{(l)}) \right)}{\partial a_i} \right|, \tag{5}$$

where $p_\theta(x^{(l)}))$ are the class probabilities. By pruning neurons based on their effect on the softmax outputs (and otherwise following Equation 3), we improve the complementarity ratio by 2.22% on CIFAR-10 Krizhevsky (2009), yielding better OOD detection performance.

## 5.2 MULTI-GUIDANCE: EXPLORING COMPLEMENTARITY AMONG HIDDEN LAYERS FOR NN GUIDANCE

So far, we have focused on improving the classifier-based score used for guidance. However, we have largely overlooked the other component of the guidance mechanism—the distance-based score computed via nearest neighbors (NN). Previous works Park et al. (2023); Lee et al. (2018) that employ distance-based scores rely exclusively on the penultimate layer of the DNN as the feature representation for NN search and distance computation.

However, different layers in a DNN capture varying levels of semantic information, which may offer complementary signals for OOD detection. For example, Figure 1 shows the distribution of activations from the last two layers of the pretrained DenseNet on CIFAR10, clearly illustrating the difference in the encoded information between the ID and OOD data.

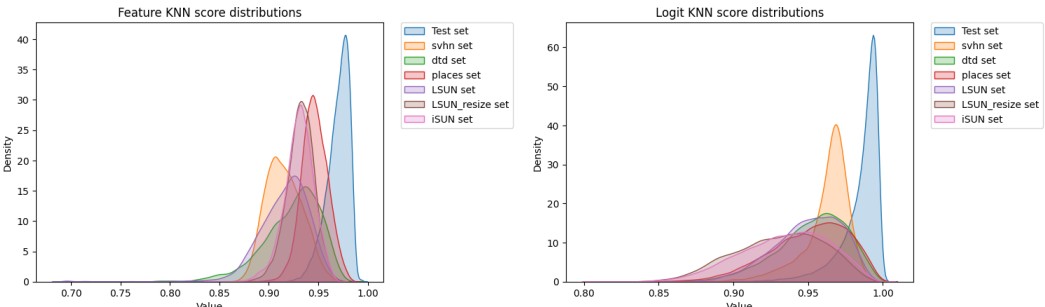

(a) The feature NN guidance score distributions obtained with the pretrained densenet on CIFAR10.

(b) The logit NN guidance score distributions obtained with the pretrained densenet on CIFAR10.

Figure 1: An illustration for the NN guidance scores on different layers using the pretrained DenseNet model on CIFAR10 dataset.

To this end, we propose *multi-guidance*—an approach that leverages the complementary information from different DNN layers to enhance the NNGuide framework. Suppose we aim to use NN scores $\{G_1, \ldots, G_n\}$, derived from intermediate DNN layers $\{l_1, \ldots, l_n\}$, to guide the classifier-based score $S$. Each $G_i$ is computed as the normalized average cosine similarity between its feature vector ($z_i$) and the ones of its top-k neighbors ($z_i^{(j)}$) in layer $l_i$. Then, the multi-guidance score is defined as:

$$G_i(\mathbf{x}) = \frac{1}{k}\sum_{j=1}^{k}\frac{1+\cos(z_i^{(j)}, z_i)}{2}, \quad G_{\text{multi}}^{\overline{1,n}} = S \cdot \prod_{i=1}^{n} G_i = G_{\text{multi}}^{\overline{1,n-1}} \cdot G_n \quad (6)$$

In other words, multi-guidance is achieved by replacing the single-layer NN-based score with a product of NN-based scores from multiple layers, as shown in the first equality.

A more informative perspective on multi-guidance stems from recognizing that guidance is simply the operation of multiplying two scores, which do not necessarily need to be a classifier- and a distance-based score. Thus, multi-guidance can be defined recursively—as expressed by the second equality in Equation 6—by treating it as NN guidance applied to a score that is itself the result of prior guidance (originating from different layers' NN scores).

This recursive formulation offers deeper insight, as it allows us to analyze multi-guidance using our complementarity ratio—by computing how complementary $G_{\text{multi}}^{\overline{1,n-1}}$ and $G_n$ are. This, in turn, enables a principled selection of DNN layers that can improve the final detection framework. Going forward, we refer to multi-guidance as $G_{\text{multi}}$, unless we want to specify the used layers explicitly.

### 5.3 CONNGUIDE: PUTTING IT ALL TOGETHER

We now bring together the components introduced above into a unified OOD detection pipeline: CoNNGuide. Given a test input $\mathbf{x}$, we first extract intermediate activations from a pre-trained classifier. These activations are pruned using our P-Shapley (see Eq 5) indices and subsequently used to compute the classifier-based score $S$.

In parallel, we compute multiple NN-based scores $\{G_{n-1}, G_n\}$ from the last two (the penultimate and logits) layers. We chose the last two layers because those have low intra-class and high inter-class variance and thus are best suited for OOD detection Masarczyk et al. (2023); Lakshminarayanan et al. (2017). These scores are combined via Equation 6 to form the multi-layer guidance score. Finally, we use the multi-guidance score—i.e., $S_{\text{CoNNGuide}} = G_{\text{multi}}$—to detect OOD examples. This design yields a simple, modular, and highly effective OOD detection system capable of achieving state-of-the-art performance.

## 6 EXPERIMENTAL RESULTS

In this section, we evaluate the performance of our strong baseline NNGuide++, and our complete method, CoNNGuide, and perform the analysis of the novel components.

**Computing resources.** All experiments were performed using the NVIDIA GeForce RTX 4080 Laptop graphics card and the CPU i9-13900HX. When performing GPU-accelerated nearest neighbour (NN) search, it takes up to 10 minutes each, depending on the model and dataset.

**Competitive Methods.** The goal of this work is not only to develop a new OOD detection approach but also to perform a thorough comparison to existing methods. We compare our method CoNNGuide against a comprehensive suite of baseline and state-of-the-art (SoTA) OOD detection techniques, including: MSP Hendrycks & Gimpel (2018), MaxLogit and KL Hendrycks et al. (2022), Energy Liu et al. (2020), ReAct Sun et al. (2021), Mahalanobis Lee et al. (2018), DICE Sun & Li (2022), KNN Sun et al. (2022), NNGuide Park et al. (2023), and LINe Ahn et al. (2023). These methods span both traditional and recent advancements in OOD detection and serve as strong baselines to assess the effectiveness of our approach.

**Evaluation Metrics.** We adopt the standard evaluation metrics commonly used in recent SoTA works Ahn et al. (2023); Sun & Li (2022); Park et al. (2023): (i) the false positive rate at 95% true positive rate (FPR95), and (ii) the area under the receiver operating characteristic curve (AUROC). Unless stated otherwise, all reported values correspond to averages over the relevant OOD datasets used in each experiment.

### 6.1 EVALUATION ON CIFAR BENCHMARKS

**Implementation Details.** We evaluate the proposed methodology on the CIFAR10 and CIFAR100 datasets Krizhevsky (2009), the common OOD detection benchmarks. Their test sets are considered as ID data. We use the same pretrained models in Sun & Li (2022): the DenseNet-101 Huang et al. (2017) being trained for 100 epochs with a learning rate of 0.1, batch size 64, weight decay of 0.0001 and momentum of 0.9. The penultimate layer (i.e., feature vectors) has a size of 342. We also chose the same OOD datasets in Sun & Li (2022) to apply a valid comparison with the competitive methods; these sets are Textures Cimpoi et al. (2014), SVHN Netzer et al. (2011), Places365 Zhou et al. (2017), LSUN-Crop Yu et al. (2016), LSUN-Resize Yu et al. (2016), and iSUNXu et al. (2015). The results are shown in Table 2.

| Method | CIFAR-10 | | CIFAR-100 | |
|---|---|---|---|---|
| | FPR95 ↓ | AUROC ↑ | FPR95 ↓ | AUROC ↑ |
| MSP | 48.6 | 92.52 | 80.75 | 73.83 |
| MaxLogit | 26.56 | 94.65 | 69.78 | 80.37 |
| KL | 26.38 | 94.76 | 69.82 | 80.7 |
| Energy | 26.38 | 94.64 | 69.84 | 80.47 |
| ReAct | 22.9 | 95.87 | 65.42 | 84.1 |
| Mahalanobis | 17.65 | 95.84 | 32.73 | 91.62 |
| DICE | 21.7 | 94.97 | 51.59 | 86.54 |
| KNN | 16.12 | 96.8 | 42.18 | 87.54 |
| DICE + ReAct | 16.11 | 96.74 | 44.53 | 87.2 |
| NNGuide | 22.36 | 95.9 | 62.16 | 81.61 |
| LINe | 14.7 | 97.13 | 31.71 | 89.21 |
| **NNGuide++** | **13.48** | **97.4** | **28.57** | **90.87** |
| **CoNNGuide (Ours)** | **12.77** | **97.45** | **26.01** | **92** |

Table 2: OOD detection performance comparison between our methods and the competitive methods on the CIFAR datasets.

**SoTA Comparison.** Table 2 demonstrates the comparisons of our methods to existing baselines. Significantly, our strong baseline - NNGuide++, already outperforms all the existing methods by a sizable margin. More precisely, the strong baseline NNGuide++ surpasses LINe, which has the best performance across the existing methods on CIFAR10 and CIFAR100, by 1.22% and 3.14% in terms of FPR95 on each dataset. On top of that, our full methods - CoNNGuide - further outperforms NNGuide++ by 0.71% and 2.56%, resulting in a total improvement of 1.93% and 5.7%. The same observations can be made for AUROC, where CoNNGuide outperforms LINe by 0.32% and 2.79% on CIFAR10 and CIFAR100 datasets. These results showed that CoNNGuide had successfully surpassed the existing SoTA methods by identifying the important feature (i.e., significant neurons in the penultimate layer) more accurately and introducing additional guidance from the logit layer.

## 6.2 EVALUATION ON IMAGENET

**Implementation Details.** The real vision classification is more complicated and possesses more target classes. To validate the OOD detection performance of our methodology in real-world scenarios, we evaluate CoNNGuide on the Imagenet-1K dataset Russakovsky et al. (2015), which is a large-scale dataset with high-resolution images. Similar to the CIFAR implementation, we utilize the test set from Imagenet as our ID data. We considered five OOD datasets: Textures Cimpoi et al. (2014), Places365 Zhou et al. (2017), iNaturalist Van Horn et al. (2018), SUN Xiao et al. (2010) and OpenImage-OWang et al. (2022), the first four datasets are called "curated datasets" and are used for the standard benchmarking on Imagenet. We use two pretrained models for the evaluation: a ResNet50 He et al. (2016) and a RegNet PyTorch Team (2024), which are both trained on Imagenet-1K. The results are displayed in Table 3.

| Method | ResNet50 | | RegNet | |
|---|---|---|---|---|
| | FPR95 ↓ | AUROC ↑ | FPR95 ↓ | AUROC ↑ |
| MSP | 65.08 | 82.75 | 45.54 | 88.27 |
| MaxLogit | 58.05 | 87 | 28.12 | 92.29 |
| KL | 57.78 | 87 | 26.8 | 92.47 |
| Energy | 57.78 | 87 | 26.77 | 92.49 |
| DICE | 35.7 | 90.9 | 26.77 | 92.49 |
| Mahalanobis | 46.34 | 90.19 | 35.32 | 92.22 |
| KNN | 53.53 | 85.2 | 25.6 | 93.08 |
| NNGuide | 26.86 | 92.69 | 17.97 | 95.42 |
| LINe | 20.7 | 95.03 | 26.58 | 92.5 |
| **NNGuide++** | **17.93** | **96.1** | **16.04** | **96.36** |
| **CoNNGuide (Ours)** | **16.36** | **96.43** | **14.85** | **96.74** |

Table 3: Performance comparison for the curated OOD sets between our methods and the competitive methods on the Imagenet dataset. The shown **FPR95** and **AUROC** are the average values over the considered OOD datasets except OpenImage-O.

**SoTA Comparison.** For the ResNet50 model, the built strong baseline NNGuide++ also outperforms the existing SoTA methods directly, and by introducing the Multi-guidance and P-Shapley indices, CoNNGuide achieves the state-of-the-art performance by surpassing LINe with an improvement of 4.34% and 1.4% in terms of FPR95 and AUROC. Same for the RegNet model, CoNNGuide outperforms NNguide, which is the current SoTA for the OOD detection on Imagenet, by reducing FPR95 to 14.85% and increasing AUROC to 96.74%. These results demonstrate that CoNNGuide possesses a good performance for OOD detection on large-scale and high-resolution scenarios, which can be seen as proof of its application in real-world situations.

| Method | PShap | MD | FPR95 (Curated) ↓ | AUROC (Curated) ↑ | FPR95 ↓ | AUROC ↑ |
|---|---|---|---|---|---|---|
| NNGuide++ | | | 17.93 | 96.1 | 22.53 | 94.89 |
| NNGuide++ with PShap | ✓ | | 17.55 | 96.2 | 22.25 | 94.87 |
| NNGuide++ with MD | | ✓ | 16.48 | 96.45 | 21.89 | 95.1 |
| CoNNGuide (Ours) | ✓ | ✓ | 16.36 | 96.43 | 21.68 | 95.12 |

Table 4: Ablation study on the proposed novel techniques with the pretrained ResNet50 on Imagenet. **FPR95 (Curated)** and **AUROC (Curated)** are the average values over the chosen OOD datasets except OpenImage-O. **FPR95** and **AUROC** are the average values over all the OOD datasets.

| Method | PShap | MD | FPR95 ↓ | AUROC ↑ |
|---|---|---|---|---|
| NNGuide++ | | | 28.58 | 90.87 |
| NNGuide++ with PShap | ✓ | | 28.49 | 91.05 |
| NNGuide++ with MD | | ✓ | 26.22 | 91.8 |
| CoNNGuide (Ours) | ✓ | ✓ | 26.01 | 92 |

Table 5: Ablation study on the proposed techniques with the pretrained DenseNet on CIFAR100.

## 6.3 ANALYSIS

The goal of this section is to evaluate individual contributions and analyze the complementarity properties of the novel components of CoNNGuide introduced in this paper.

**Ablation Study.** We analyze the individual components by performing ablations on (i) the P-Shapely indices used to prune the activation and weights of the network, and (ii) the multi-guidance approach, gathering information from multiple DNN layers to improve OOD detection. Since both components are built on top of our strong baseline, NNGuide++, we chose it as the baseline here.

Table 5 and 4 show the ablation study for the main CoNNGuide's components. The results in both tables show that each novel component improved the OOD detection performance. More precisely, for the Imagenet dataset, introducing the P-Shapley indices for activation importance identification reduces FPR95 by $0.38\%$ and the NN guidance through logit information improves FPR95 by $1.45\%$. With both techniques, we successfully reduce FPR95 to $16.36\%$. Similarly, for the CIFAR100 dataset, the P-Shapley indices and the logit NN guidance reduce FPR95 by $2.57\%$ compared to the proposed strong baseline NNGuide++, and improve the AUROC by $1.13\%$.

| Base scores | Complementarity Ratios (%) | FPR95 ↓ | AUROC ↑ |
|---|---|---|---|
| Energy | 89.26 | 24.79 | 94.99 |
| ReAct | 89.49 | 21.21 | 96.13 |
| ReAct+PShap | 91.71 | 13.32 | 97.44 |
| ReAct+PShap+MD | 91.92 | 12.77 | 97.45 |

Table 6: Complementarity ratios and performance after the feature guidance for the detected OOD examples with the feature NN scores and different base scores on CIFAR10 dataset. PShap stand for P-Shapley and MD is Multi-guidance. The OOD score used in ReAct is Energy.

**Complementarity of CoNNGuide Components.** Here, we measure the complementarity properties of the components of CoNNGuide. To do so, we build up the CoNNGuide pipeline, starting from vanilla Energy scoreLiu et al. (2020), and measure the complementarity of the classification- and NN-based scores on CIFAR10. As shown in Table 6, both the selection of important neurons based on P-Shapley indices and Multi-guidance improve the complementarity score and lead to the improved final performance of CoNNGuide.

## 7 CONCLUSION

In this work, we revisited the problem of OOD detection through the lens of nearest-neighbor guidance. We introduced a principled framework for understanding and exploiting the complementarity of OOD detection scores. By carefully analyzing the conditions under which classifier-based and distance-based scores synergize, we proposed a systematic method to construct strong baselines from existing components. We further enhanced this foundation with two novel contributions: P-Shapley pruning, which improves the robustness and complementarity of classifier-based scores, and multi-guidance, which aggregates signals across multiple feature layers. These insights culminated in CoNNGuide, a simple yet highly effective detection pipeline that achieves state-of-the-art results across CIFAR and ImageNet benchmarks. It must be noted that our method inherits the limitations of NNGuide, and, for example, is expected to benefit from combining multiple NN-based scores to a lesser degree. Nevertheless, our findings highlight the central role of complementarity in guided OOD detection and offer the potential to build upon this work in future research.

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
