# OpenReview forum: "Complementarity Matters: A Closer Look at Nearest Neighbor Guidance for OOD Detection"
_ICLR.cc/2026/Conference — ICLR 2026 Conference Withdrawn Submission_

### Official Review · Reviewer_8KjP · 2025-10-22

**Soundness:** 2
**Presentation:** 1
**Contribution:** 2
**Rating:** 2
**Confidence:** 5

**Summary:**

Regarding the out-of-distribution (OOD) detection method, this work proposes the complementarity issue for the effectiveness of the NNGuide method, where Energy and nearest neighbor are combined. The key observation lies in that Energy and nearest neighbor can detect different OOD samples, thereby their combination is a kind of complementarity, enhancing detection results. Based on this observation, a strong baseline method is further proposed, including model pruning and guidance from hidden layers. Empirical results validate the effectiveness of the proposed method.

**Strengths:**

1.	The proposed complementarity issue, defined in Eqn.(4), is interesting and potentially reasonable for the effectiveness of ensembling different OOD detection scores.
2.	Experiments cover both small-scale and large-scale datasets.

**Weaknesses:**

1.	The formatting of all in-text citations is non-standard, as the outermost parentheses are missing. This non-standard formatting significantly disrupts the reading flow (have the authors read the paper themselves?). All references should be revised into a correct format.

2.	The writing of this work should be deeply revised. The reviewer lists a few suggestions for reference. This work, in its current shape, looks like an incremental work based on an existing NNGuide method.

2.1.	Firs of all, NNGuide, published in 2023, is obviously not the current SOTA detection method. Besides, the proposed complementarity issue should be extended to a series of detection methods that ensemble different types of detection scores, forming a general framework, instead of just being applied to NNGuide.

2.2.	The introduction of the network pruning strategy is not convincing nor fluent. I believe there must be other techniques that can achieve the complementarity. Why the network pruning is selected?

2.3.	In Sec.4, given the Eqn.(4) defining the complementarity, the next step should be analyzing the NNGuide using Eqn.(4), in order to validate the key insight of this work. However, there are no analysis after Eqn.(4), and a new enhanced NNGuide++ is immediately proposed, which is confusing.

2.4.	More in-depth explorations are essential. The complementarity remains a superficial metric that only tells the diversities in detected OOD samples between two methods. But why the two methods can detect different samples are not clearly explained. Investigations towards this direction would further strengthen this work.

3.	In the compared baselines in experiments, the most recent one was published in 2023. Therefore, the authors are suggested to incorporate more latest detection methods. For example, the ensemble of the distance-based Kernel PCA and the logits-based Energy is investigated in [a], which could also be a suitable instance for the complementarity to be exploited.

[a] Kernel PCA for Out-of-Distribution Detection. NeurIPS 2024.

**Questions:**

My questions are from some weaknesses above.
4.	Why the network pruning is selected to enhance NNGuide? It is inappropriate to introduce something new without solid and convincing explanations.
5.	What are the potential reasons that cause such diversities in detected samples between two methods? Can the authors provide discussion into examples that two detection methods lead to terrible complementarity?

---

### Official Review · Reviewer_3kcE · 2025-10-22

**Soundness:** 3
**Presentation:** 2
**Contribution:** 2
**Rating:** 2
**Confidence:** 4

**Summary:**

The paper revisits the nearest-neighbor guidance for OOD detection and argues its gains come from complementarity between classifier based and NN based scores. It (i) quantifies complementarity via a union style ratio, (ii) builds a stronger NNGuide++ baseline (Energy + ReAct + pruning), (iii) replaces logit based Shapley with probability aware pruning, and proposes multi-guidance that multiplies NN guidance from both penultimate and logit layers. CoNNGuide reports improved FPR95 on CIFAR10/100 and ImageNet benchmarks using Densenet, ResNet50 and RegNet architectures.

**Strengths:**

- **Easy to follow.** The paper is clear with a simple pipeline view and stepwise ablations.

- **Constructive combination.** It recognizes complementary strengths/weaknesses of classifier and distance based scores and combines them into a unified score.

- **Strong reported results.** CoNNGuide improves over NNGuide on CIFAR and ImageNet benchmarks.

- **Linking the metric with the performance.** The observation that complementarity ratio correlates with improvements after guidance is valuable and it gives an interpretable knob for the design.

**Weaknesses:**

Most of the weaknesses I would raise is going to be about the evaluation protocol used in the paper. I am not against the intrinsic value of the suggested score but I do believe that the method is not stress tested enough to validate the claims.

- **Evaluation falls short.** The empirical scope is narrow for today's OOD landscape. Although authors mention the near/far OOD characteristics of classifier and distance based methods, no tests on standard near-OOD suites (e..g NINCO/SSB-Hard). To substantiate the claims on line 49, the paper needs:
  - A table directly comparing distance vs classifier based methods on near and far OOD,
  - CoNNGuide's numbers on both regimes to support it combines the strengths of the both.

- **Outdated backbones.** Experiments focus on DenseNet101, ResNet50 and RegNet; modern backbones (ConvNeXt, DeiT, Swin, EVA) and their pretrained variants when available are used in the recent works [1], [2], [3] to address the performance variance coming from different backbones. Moreover, there are works who demonstrate their robustness to different learning paradigms (using models trained with SupConLoss/CLIP losses).

- **Baselines missing.** Simple, strong baselines [1], [2] (includes and compares many methods), [3], [4], [5], [6], [7] are not included, limiting comparative insight and related work discussion.

- **Hyperparameters and their presentation.** CoNNGuide combines multiple methods therefore inherits their hyperparameters (5 in total). The main paper does not foreground this and no sensitivity analysis is provided.

- **Benchmarking protocol deviations.** CIFAR uses LSUN crop and LSUN resize which are very easy datasets that can inflate the averages. I do not think using both of them is the standard evaluation for CIFAR anymore. In addition, ImageNet evaluation deviates from standard OpenOOD protocols, which raises credibility concerns.

**Questions:**

- Can the authors relate angle-based methods in [2] and [3] to the distance based ones? Do authors observe any differences in performance when tested on near vs far OOD benchmarks?

- The argument in 59-60, claiming that putting two methods would result in a better performance is not trivial. It makes the claim with a general language as if when we take any two scores with complementary characteristics, the guided combination will result in a better score. I do not find this very trivial as it will largely depend on the guidance accuracy.

- Line 62 claims different hidden layers produce features with different properties. This part needs either a citation to a work that studies OOD performance of different layer representations, or an empirical demonstration of the phenomena that is claimed.

- I could not find in-depth explanations of the hyperparameters. Please introduce them in the equations and properly define them.

## References
[1] Mueller & Hein, 2025. *Mahalanobis++: Improving OOD Detection via Feature Normalization*. arXiv:2505.18032.
[2] Bitterwolf, Mueller, Hein, 2023. *In or out? Fixing ImageNet OOD detection evaluation*. arXiv:2306.00826.
[3] Demirel, Fumero, Locatello, 2024. *Out-of-Distribution Detection with Relative Angles*. arXiv:2410.04525.
[4] Ren et al., 2021. *A simple fix to Mahalanobis distance for improving near-OOD detection*. arXiv:2106.09022.
[5] Liu & Qin, 2025. *Detecting OOD through the lens of neural collapse*. CVPR.
[6] Ammar et al., 2023. *NECO: Neural collapse based OOD detection*. arXiv:2310.06823.
[7] Liu, L., & Qin, Y. (2023). Fast decision boundary based out-of-distribution detector. arXiv preprint arXiv:2312.11536.

---

### Official Review · Reviewer_zEdT · 2025-10-30

**Soundness:** 3
**Presentation:** 3
**Contribution:** 3
**Rating:** 4
**Confidence:** 3

**Summary:**

This paper examines the prevailing score-based approach to OOD detection. Prior work shows that combining classifier-based and nearest-neighbor (NN) based scores, often yields the strongest detectors, but the reason for its success has been unclear. This work analyze NN guidance and identify its core driver: complementarity between the two scores. When each score flags different OOD examples, their combination improves detection. Building on this, this work (i) develop a strong NN-guidance baseline that increases score complementarity, (ii) introduce a probability-aware pruning strategy that further enhances complementarity and accuracy, and (iii) propose a multi-layer fusion that aggregates complementary signals from different hidden layers. Integrating these components into a unified framework that achieves superior results on CIFAR and ImageNet.

**Strengths:**

1. The paper targets an important direction in OOD detection and clearly identifies complementarity between classifier and NN scores, which clarifies when and why NN guidance helps.

2. It establishes strong baselines with fair comparisons to prior post-hoc methods. The proposed approach is model-agnostic, training-free, and technically sound, making it practically valuable.

3. Extensive experiments across multiple architectures and datasets demonstrate strong performance and robustness.

**Weaknesses:**

1. Fusion rule feels ad hoc. Consider an adaptive or lightly learned combiner and report robustness to score rescaling and noise.

2. Provide a systematic analysis for multi-layer guidance: show a layer-by-layer complementarity matrix and report performance for the top-r complementary layers to justify the final selection.

3. Evaluate on modern architectures (e.g., ViT, CLIP encoders with zero-shot classification) to confirm whether the observed trends hold beyond CNNs.

4. Hyperparameter selection lacks guidance. Propose a stylized procedure (e.g., select k via ID cross-validation optimizing AUROC proxy) and include sensitivity plots.

**Questions:**

1. Provide a fine-grained analysis of layer selection for multi-guidance: which layers yield the most complementary signals, and do the best layers vary across datasets?

2. Examine whether the observed complementarity patterns persist across architectures, including modern models such as ViT, CLIP encoders.

3. Clarify potential overlap between backbone pretraining data and OOD test sets; confirm results on curated, non-overlapping OOD sets.

4. Include a hyperparameter selection recipe, including for choosing k, with sensitivity analyses.

---

### Official Review · Reviewer_fxgh · 2025-11-01

**Soundness:** 2
**Presentation:** 3
**Contribution:** 2
**Rating:** 4
**Confidence:** 3

**Summary:**

This paper analyzes why nearest-neighbor guidance (NNGuide) improves OOD detection and identifies score complementarity as the key factor. Based on this insight, the authors propose NNGuide++, P-Shapley pruning, and multi-guidance, which are combined into CoNNGuide, a unified framework achieving state-of-the-art performance on CIFAR and ImageNet benchmarks.

**Strengths:**

- Introduces the novel concept of score complementarity to explain NNGuide’s effectiveness.
Modular, practical, and training-free method.
- Strong empirical results with comprehensive ablation studies.
- Outperforms prior state-of-the-art methods on multiple benchmarks.

**Weaknesses:**

1. **Limited Novelty in Components**:
While the overall framework is well-designed, most components (e.g., energy-based scores, pruning, cosine similarity for NN search) are based on existing methods. The main novelty lies in combining them effectively rather than proposing fundamentally new algorithms.

2. **Lack of Theoretical Justification for Complementarity**:
The paper introduces "score complementarity" as a key insight, but this concept is only supported by empirical correlation. There is no formal analysis or theoretical framework to understand when and why complementarity leads to improved performance.

3. **Scalability Concerns**:
The method relies on nearest-neighbor searches over a feature bank, which may become computationally expensive or memory-intensive on large-scale datasets or in real-time applications. While GPU acceleration is mentioned, practical efficiency in large deployments remains unclear.

4. **No Specific Analysis on Near- vs. Far-OOD**:
The paper claims that classifier-based and distance-based scores are better at near- and far-OOD detection respectively, but does not include experiments or metrics that separately evaluate performance on these two types of OOD samples. Such analysis would strengthen the paper’s core argument about score complementarity.

**Questions:**

Refer to the weakness section.

**Details Of Ethics Concerns:**

No Ethics Concerns.

---

### Note · Authors · 2025-11-29

**Comment:**

We would like to further improve the paper to obtain more promising results.

**Withdrawal Confirmation:**

I have read and agree with the venue's withdrawal policy on behalf of myself and my co-authors.